# Polish and New Zealand Propolis as Sources of Antioxidant Compounds Inhibit Glioblastoma (T98G, LN-18) Cell Lines and Astrocytoma Cells Derived from Patient

**DOI:** 10.3390/antiox11071305

**Published:** 2022-06-29

**Authors:** Justyna Moskwa, Sylwia Katarzyna Naliwajko, Renata Markiewicz-Żukowska, Krystyna Joanna Gromkowska-Kępka, Jolanta Soroczyńska, Anna Puścion-Jakubik, Maria Halina Borawska, Valery Isidorov, Katarzyna Socha

**Affiliations:** 1Department of Bromatology, Faculty of Pharmacy with Division of Laboratory Medicine, Medical University of Białystok, Mickiewicza 2D Street, 15-222 Białystok, Poland; sylwia.naliwajko@umb.edu.pl (S.K.N.); renmar@poczta.onet.pl (R.M.-Ż.); krystyna.gromkowska-kepka@umb.edu.pl (K.J.G.-K.); jolanta.soroczynska@umb.edu.pl (J.S.); anna.puscion-jakubik@umb.edu.pl (A.P.-J.); borawska@umb.edu.pl (M.H.B.); katarzyna.socha@umb.edu.pl (K.S.); 2Forest Faculty, Białystok University of Technology, Wiejska 45A Street, 15-351 Białystok, Poland; v.isidorov@pb.edu.pl

**Keywords:** propolis, polyphenols content, glioma cells, cancer prevention and treatment

## Abstract

Gliomas, including glioblastoma multiforme and astrocytoma, are common brain cancers in adults. Propolis is a natural product containing many active ingredients. The aim of this study was to compare the chemical composition, total phenolic content and concentration of toxic elements as well as the anticancer potential of Polish (PPE) and New Zealand (Manuka—MPE) propolis extracts on diffuse astrocytoma derived from patient (DASC) and glioblastoma (T98G, LN-18) cell lines. The antioxidants such as flavonoids and chalcones (pinocembrin, pinobanksin, pinobanksin 3-acetate and chrysin) were the main components in both types of propolis. The content of arsenic (As) and lead (Pb) in MPE was higher than PPE. The anti-proliferative study showed strong activity of PPE and MPE propolis on DASC, T98G, and LN-18 cells by apoptosis induction, cell cycle arrest and attenuated migration. These findings suggest that despite their different geographic origins, Polish and New Zealand propolis are sources of antioxidant compounds and show similar activity and a promising anti-glioma potential in in vitro study. However, further in vivo studies are required in order to assess therapeutic potential of propolis.

## 1. Introduction

Gliomas are tumors of neuroepithelial origin and represent approximately 40% of primary intracranial tumors. Diffuse astrocytomas belong to a category of diffuse gliomas which arise from glial cells. A glioma is a slow-growing brain tumor and tends to grow into and infiltrate neighboring, healthy tissue brain. Glioblastoma multiforme (GBM) is the most aggressive malignant tumor in the CNS and has a poor prognosis [1]. A characteristic feature of glioma is the diversity of histological features and cell composition. Currently, the main method of treatment this type of tumor is surgery, which offers rapid relief from the symptoms of high intracranial pressure and provides a chance to remove or reduce neurological defects. The next step is radio and chemotherapy [2]. However, patients with the GBM treated with radiotherapy combined with temozolomide (TZM) expect a median survival of only 15 months [3]. Therefore, natural compounds which could enhance currently available treatment modalities are sorely needed.

Propolis is a natural product composed of tree and plant resin, bee wax, pollen and gland secretions of bees. When compared to other natural products, propolis is unique, since it is of both plant and animal origin. It contains a wide range of active components, whose concentrations depends primarily on the geographical provenance, season of the year, and the breed of bees. There are several types of propolis: “Poplar” (European, Chinese, North and South American, including Manuka propolis from New Zealand), “Brazilian green” (containing artepillin-C), “Red” (from Cuba, Brazil, Mexico), “Birch” (from Russia), “Mediterranean” (Greece, Crete, Sicily, Malta), “Pacific” (from Okinawa, Taiwan, Indonesia) and “Clusia” (from Cuba and Venezuela) [4]. Hence, various biological activities of propolis have been reported by many authors. The most active compounds are flavonoids (e.g., chrysin, apigenin, pinocembrin, pinobanksin, kaempferol), aromatic acids (e.g., p-coumaric, ferulic), and esters (caffeic acid phenethyl ester—CAPE) [5,6]. A number of studies concerning the anti-tumor activity of propolis on various cancer cell lines such as human colorectal cancer (DLD-1) [4], human lung cancer (A549) [7], gastric cancer (HGC27) [8], and human prostate cancer (PC3) [9] have been published. The chemical composition and antiproliferative effect of propolis from Poland on the human glioblastoma multiforme cell line U87MG has been confirmed in our previous studies [5,10,11]. The research studies have focused on the potential utilization of propolis phenolic compounds in the development of new anti-cancer drugs [12,13]. The role of antioxidant action in cancer cells is complex and not completely understood. Scientific research shows that antioxidants are able to decrease the tumor formation risk by preventing ROS-induced oxidation of DNA and sub-sequent DNA damage [14], but on the other hand, Schafer et al. showed that antioxidant activity may promote the survival of preinitiated tumor cells in unnatural matrix environments and thus enhance malignancy [15].

It is well known that propolis has a very rich chemical composition, and its compounds show a multidirectional effect on the human body. The present study compare the antiproliferative activity of propolis from Poland and from New Zealand on different types of brain tumor—human diffuse astrocytoma cell line (DASC) derived from a patient with Grade II glioma and glioblastoma multiforme T98G and LN-18 cell lines.

## 2. Materials and Methods

### 2.1. Materials

DMEM/Ham’s F12 with L-glutamine was purchased from PAA Laboratories GmbH (Pasching, Austria). Dulbecco’s modified eagle medium (DMEM), fetal bovine serum (FBS), minimal essential medium eagle (MEM) with L-glutamine, trypsin-EDTA, penicillin, streptomycin were purchased from Gibco (Thermo Fisher Scientific, Waltham, MA, USA). Calcium-free phosphate-buffered saline (PBS) was received from Biomed (Lublin, Poland). Bis(trimethylsilyl)trifluoroacetamide (BSTFA) with an addition of 1% trimethylchlorosilane, C10–C40 n-alkane standard solution, methylthiazolyl diphenyl-tetrazolium bromide (MTT), dimethyl sulfoxide (DMSO), pyridine, trichloroacetic acid, and trizma base were obtained from Sigma-Aldrich (St. Louis, MO, USA). Ethanol at 95% was obtained (AWW Group, Poland). The scintillation cocktail was purchased from PerkinElmer (Boston, MA, USA) and methyl-3H thymidine from MP Biomedicals, Inc. (Irvine, CA, USA).

### 2.2. Sample Preparations

Propolis of *Apis mellifera* was collected in the Podlasie region (northeastern Poland). To prepare the ethanolic extract of Polish propolis (PPE), 20 g of crushed propolis was extracted on a shaker with 80 g of 70% ethanol for 12 h in a darkened place. The extract was centrifuged at 2500 rpm for 10 min at 20 °C, evaporated (40 °C) in a rotary evaporator (Rotavapor R-3, Buchi, Switzerland) and lyophilized. The dry Polish propolis extract (PPE) was protected from light and kept frozen at −20 °C. The yield of the prepared extracts (% *w*/*w*) in terms of the starting material was 47.6.

Propolis Manuka Health New Zealand (Bio 30) ethanolic tincture was purchased from the manufacturer. The tincture was evaporated (40 °C) in a rotary evaporator (Rotavapor R-3, Buchi, Switzerland) and lyophilized. The dry Manuka propolis extract (MPE) was protected from light and kept frozen at −20 °C.

The extracts were dissolved in DMSO and prepared as 1 mg/mL stock solution (calculated as dry extracts) in the culture medium.

### 2.3. Gas Chromatography–Mass Spectrometry (GC-MS) Analysis

At this stage, 5 mg of PPE and MPE were diluted with 220 μL of pyridine and 80 μL of BSTFA with an addition of 1% trimethylchlorosilane. The reaction mixture was sealed and heated for 0.5 h at 60 °C to form trimethylsilyl (TMS) derivatives.

GC-MS analyses of PPE and MPE were performed using GC-MS on an HP 6890 gas chromatograph with a mass selective detector MSD 5973 (Agilent Technologies, Santa Clara, CA, USA) equipped with a ZB-5MSi fused silica column (30 m, 0.25 mm i.d., 0.25 μm film thickness), with electronic pressure control and a split/splitless injector. Helium flow rate through the column was 1 mL/min in a constant flow mode. The injector worked at 250 °C in the split (1:50) mode. The initial column temperature was 50 °C, rising to 310 °C at 5 °C/min and the higher temperature was maintained for 15 min. MSD detector acquisition parameters were as follows: transfer line temperature 280 °C, MS Source temperature 230 °C and MS Quad temperature 150 °C. The EIMS spectra were obtained at the ionization energy of 70 eV. The MSD was set to scan 41–600 a.m.u. Following the integration, the fraction of each component in the total ion current was calculated. Hexane solutions of C_10_–C_40_
*n*-alkanes were separated under the above conditions. Gas chromatographic linear programmed retention indices (***I*_T_**) were calculated on the basis of the retention times of the *n*-alkanes hexane solution and separated components of the extract samples.

To identify the separated components, two independent analytical parameters were used: mass spectra and calculated retention indices. The mass spectrometric identification of non-derivatized components was performed with an automatic system for GC-MS data processing supplied by the NIST 14 library (NIST/EPA/NIH Library of Electron Ionization Mass Spectra). The mass spectra and retention indices of the components registered in the form of TMS derivatives were compared with those presented in a recently published database [16] and a private mass spectra library. Identification was considered reliable if the results of the computer search of the mass spectra library were confirmed by experimental *RI* values, i.e., if their deviation from the published database values did not exceed ± 10 u.i. (the average quantity of inter-laboratory deviation for non-polar stationary phases).

### 2.4. Total Phenolic Content Analysis

Total phenolic content (TPC) was measured using the Folin–Ciocalteu colorimetric method (FC). Absorbance versus a prepared blank was read at 760 nm using Cintra 3030 (GBC Scientific Equipment, Melbourne, Australia). The results were expressed as milligrams of gallic acid equivalent (GAE) per gram of a dry extract. The concentration of samples equaled 2 mg/mL (extract dissolved in 70% ethanol). Data were expressed as mean ± SD.

### 2.5. Toxic Elements Analysis (Arsenic, Cadmium, and Lead)

Coupled plasma mass spectrometry (ICP-MS, NexION 300D, PerkinElmer, USA) was applied to determine toxic element. Before analysis, propolis samples were mineralized according to a procedure proposed by Bielecka et al. [17]. A kinetic energy discrimination (KED) chamber was used in the case of As and the standard mode in the case of Cd and Pb. In order to correct for polyatomic interference in this configuration, kinetic energy discriminations and collisions were applied. The results were obtained in counts per second (cps) and based on calibration curves, were converted into concentrations. To determine the limit of detection (LOD), 10 independent blank determinations were made. A three-fold standard deviation (SD) from the mean value determined in concentration units was taken as the LOD. The LOD values were 0.018 μg/kg for As, 0.017 μg/kg for Cd, and 0.16 μg/kg for Pb. ICP-MS conditions for As, Cd, and Pb determination were described in our previous publication [17]. Quality control was performed by analyzing certified reference material (corn flour INCT-CF-3, Institute of Nuclear Chemistry and Technology, Warsaw, Poland) prior to the start of the analysis. The results of the quality control are summarized in Table 1.

### 2.6. Cell Culture

The study was performed using Diffuse Astrocytoma Stem-like Cells (DASC) and glioblastoma multiforme cell lines (T98G and LN-18). The DASC cell line was derived from a 43-year-old patient with diffuse astrocytoma (Grade II), as described in our previous research [18]. The study was approved by the local Ethics Committee [18]. T98G and LN-18 had been obtained from the American Type Culture Collection (ATCC, Rockville, MD, USA). The cells were cultured in a humidified incubator at 37 °C and 5% CO_2_ atmosphere, in MEM (DASC and T98G) or DMEM (LN-18) supplemented with 10% heat inactivated FBS; 100 U/mL penicillin and 0.1 mg/mL streptomycin. Subconfluent cells were detached with a trypsin-EDTA solution in PBS and counted in a Neubauer hemocytometer. Assays were performed in triplicate.

### 2.7. Cell Viability Assay 

Cell viability was measured using an MTT assay, as previously described for glioma cells [19]. The effects of PPE and MPE extracts on DASC, T98G and LN-18 cell lines were studied after 24 h, 48 h and 72 h of the treatment. The cells were cultured as follows: in a humidified incubator at 37 °C and 5% CO_2_ atmosphere; in MEM or DMEM supplemented with 10% heat inactivated FBS; with 100 U/mL penicillin and 0.1 mg/mL streptomycin. Doses of propolis (10, 20, 30, 50, 100 µg/mL) were selected in our previous experiments [11]. Cells at a density of 1 × 105 cells/mL were seeded onto 96-well plates at a volume of 200 µL per well and grown for 22 h at 37 °C in a humidified 5% CO_2_ incubator. The data were expressed as a percentage of the control (0.1% DMSO).

### 2.8. DNA Synthesis Assay

At this stage, [^3^H]-thymidine assays were performed to study DNA synthesis in the cells after the treatment. The cells were seeded (1.5 × 10^5^ cell/well) on 24-well plates in MEM or DMEM supplemented with 10% heat inactivated FBS, 100 U/mL penicillin and 0.1 mg/mL streptomycin, and exposed to the treatment medium containing DMSO (0.1%-control), PPE and MPE (30 µg/mL). The cells were cultured for 20, 44 and 68 h prior to adding 0.5 µCi of [^3^H]-thymidine per well. After 4 h of incubation with [^3^H]-thymidine, the medium was removed and the cells were washed twice with cold 0.05 M Tris-HCl and 5% trichloroacetic acid, then scraped and transferred to a scintillation cocktail. The level of [^3^H]-thymidine incorporated in the newly synthesized DNA strand was assessed by a scintillation counter in relation to the DNA synthesis in the control cells. Amount of incorporated [^3^H]-thymidine indicates the ability of cells proliferation.

### 2.9. Migration Assay (Scratch Assay)

For the scratch test, the DASC, T98G and LN-18 cells were cultured (0.5 × 10^6^ cell/well) on 6-well plates, at 37 °C in a humidified atmosphere of 5% CO_2_. After reaching 80–90% confluence, the cells in the well plates were scratched with a sterile 20 μL micropipette tip to the same length and width. After each well had been washed with PBS to remove debris, the cells were treated with PPE and MPE (30 µg/mL) and medium containing DMSO (0.1%, control), and then incubated for 42 h. The images of each treatment well were captured at 100× magnification, using an Olympus CKX 41 microscope and KcJunior program at each time point (0, 18, 42 h) and combined into one figure. The images acquired for each sample at different times were quantitatively analyzed using ImageJ 1.52v analysis software, a free image-processing and analysis program. The cell migration was calculated as a percentage of scratch area.

### 2.10. Cell Cycle Assay

The effect of PPE and MPE on the cell cycle was analyzed by the Advanced Image Cytometer NucleoCounter NC-3000 (ChemoMetec, Lillerød, Denmark), as described in our previously published study [19]. The DASC, T98G, and LN-18 cells were seeded into 6-well plates at a density of 1 × 10^6^ cells per well. After 24 h of incubation, the cells were treated with PPE and MPE (30 µg/mL) or a medium containing DMSO (0.1%, control). After 24 h of cell treatment, the test was performed using 1–2 × 10^6^ cells, according to the 2-step cell cycle assay protocol of the manufacturer (ChemoMetec, Lillerød, Denmark). The results are presented as the percentages of the cells in different cell cycle phases: subG1, G1/G0, S, and G2/M.

### 2.11. Annexin V Assay

Using image analysis, the NucleoCounter^®^ NC-3000™(ChemoMetec, Lillerød, Denmark), we indicated a quantification of early apoptotic cells based on Annexin V binding and PI exclusion. Cells (2−4 × 10^5^) were stained with Annexin V-CF488A conjugate along with Hoechst 33342. Just before analysis, cells were mixed with PI to stain nonviable cells. The DASC, T98G, and LN-18 cells were seeded into 6-well plates at a density of 1 × 10^5^ cells per well, and after 24 h of incubation, they were treated with PPE and MPE (30 µg/mL). After 48 h of incubation with the studied agents, the assay was performed following the manufacturer’s protocol for the Annexin V assay (ChemoMetec, Lillerød, Denmark).

### 2.12. Statistical Analysis

All data were analyzed using Statistica software, version 13.3. The results were expressed as mean ± SD and statistically compared to the control. Values were tested for a normal distribution using the Shapiro–Wilk test. Differences between two groups were analyzed using Student’s *t*-test or Mann–Whitney U test. *p* < 0.05 was considered to be statistically significant.

## 3. Results

### 3.1. Chemical Composition and Total Phenolic Content of PPE and MPE

In this study, more than 100 individual compounds in PPE and more than 150 compounds in MPE were identified by GC-MS analysis (Appendix A). A list of the main constituents is presented in Table 2, where both propolis extracts contained a lot of antioxidants compounds. Flavonoids and chalcones were the main components of both examined types of propolis (PPE, 49.4%; MPE, 52.1%) (Table 3). The main representatives of this group of compounds in PPE and MPE were pinocembrin (8.16% and 14.64%), pinobanksin (4.25% and 4.70%), pinobanksin 3-acetate (11.27% and 9.21%), chrysin (5.33% and 5.73%), galangin (8.95% and 9.60%), respectively, and their derivatives (Table 2). Cinnamic acid derivatives such as esters 3-methyl-2-bytenyl (E)-caffeate, benzyl (E)-caffeate, benzyl (E)-p-coumarate, 2-phenylethyl p-coumarate, benzyl (E)-ferulate, CAPE, cinnamyl (E)-p-coumarate and others were the second significant group of compounds in PPE and MPE (19.8% and 14.5%, respectively) (Table 2). Considerable quantities of aromatic acids were present in both studied propolis extracts, although propolis from Poland (PPE–18.3%) contained twice as great a quantity of aromatic acids as propolis from New Zealand (MPE–7.8%) (Table 3). The main representatives of this group were p-coumaric acid, (E)-ferulic acid and (E)-caffeic acid. PPE contained high levels of p-coumaric acid (9.80%) (Table 2). TPC determination confirmed that PPE and MPE are rich sources of polyphenolic antioxidants—243.7 ± 9.0 in PPE and 245.6 ± 5.9 mg GAE/g in MPE (Table 4).

### 3.2. Toxic Elements Content

In this study, we determined the arsenic, cadmium, and lead content in Polish and Manuka propolis by ICP-MS method. The results are presented in Table 4.

### 3.3. Cell Viability

In this study, the impact on the viability was determined using different types of brain cancer cells—astrocytoma cell line derived from a patient (DASC) and two glioblastoma T98G and LN-18 cell lines from the ATCC. Dose- and time-dependent decreases in the viability (by MTT) of DASC were observed after 24, 48 and 72 h of incubation with both PPE and MPE (compared to the control) (Figure 1), and were comparable for both kinds of propolis. For the DASC cell line, we observed a significant reduction in cell numbers (*p* < 0.05) in all concentrations after 24, 48, and 72 h; for the dose of 30 µg/mL, it was 77.9 ± 4.3% and 81.3 ± 4.0% after 24 h, 58.6 ± 0.3% and 63.4 ± 7.8% after 48 h, and 47.0 ± 3.2% and 51.6 ± 8.1% after 72 h for PPE and MPE, respectively (Figure 1A–C). A significant (although lower than 10%) difference (*p* < 0.05) in the reduction in DASC cells treated with PPE in comparison to those treated with MPE was observed for the 100 µg/mL concentration after 48 h (approximately 7%) (Figure 1B) and for 20, 50, and 100 µg/mL concentrations after 72 h (8.4%, 6.9%, and 3.0%, respectively) (Figure 1C). For the T98G cell line, we observed a stronger, more significant reduction in cell numbers (*p* < 0.05) in all concentrations after 24, 48 and 72 h than for the DASC cell line; for the dose of 30 µg/mL, it was 78.4 ± 3.0% and 75.2 ± 2.3% after 24 h, 62.8 ± 1.3% and 50.8 ± 7.2% after 48 h, and 30.7 ± 7.7% and 22.0 ± 8.3% after 72 h for PPE and MPE, respectively (Figure 1D–F). Interestingly, dose-dependent decreases in the viability of T98G cells were observed after 24, 48 and 72 h, but only for the 10–50 µg/mL dose range. After the treatment of the 100 µg/mL dose, the decrease in viability was smaller than for the 50 µg/mL dose. This effect can be connected with the impact of the phytochemicals from propolis on activity of succinate dehydrogenase; however, some studies suggest that natural antioxidants may have a direct reductive potential and can interfere with MTT [20,21,22]. Therefore, for further study, a lower dose (30 ug/mL) of PPE and MPE has been used. A significant difference (*p* < 0.05) in the reduction in T98G cells treated with PPE in comparison to those treated with MPE was observed for the 50 µg/mL concentration after 24 h (Figure 1D), for 10, 20, 30, and 50 µg/mL concentrations after 48 h (Figure 1E), as well as and 20, 50, and 100 µg/mL concentrations after 72 h (Figure 1F). A significant (*p* < 0.05) reduction in cell number was observed for LN-18 in all concentrations of PPE and MPE after 24, 48 and 72 h. For the dose of 30 µg/mL, it was 81.6 ± 3.3% and 83.2 ±0.9% after 24 h, 49.1 ± 7.8% and 65.7 ± 8.0% after 48 h, 40.8 ± 2.5% and 41.1 ± 2.9% after 72 h for PPE and MPE, respectively. A significant difference (*p* < 0.05) in the reduction in LN-18 cells treated with PPE, as compared with those treated with MPE was observed for the 10, 30, 50, and 100 µg/mL concentrations after 48 h (Figure 1H), as well as for 50, and 100 µg/mL concentrations after 48 h (Figure 1I). Interestingly, PPE decreases the viability of LN-18 cells significantly more strongly than MPE.

### 3.4. DNA Biosynthesis

The impact of PPE and MPE on DNA biosynthesis in the [^3^H]-thymidine incorporation assay was examined in order to confirm if the inhibition of cell viability was caused by a reduction in proliferation capacity. As regards the DASC cell line, we found that both PPE and MPE significantly inhibited proliferation―by approximately 10.2% and 13.2% after 48 h and by approximately 23.1% and 18.6% after 72 h, respectively (Figure 2A–C). For the T98G cell line, we observed a significant reduction in proliferation capacity (*p* < 0.05) only in the case of MPE: 18.4% after 24 h, 18.6% after 48 h and 39.6% after 72 h (Figure 2D–F). For the LN-18 cell line, we found a significant reduction in proliferation capacity (*p* < 0.05) with both PPE and MPE after 24, 48, and 72 h: approximately 40.6% and 44.5% after 24 h, 39.4% and 43.3% after 48 h, and 67.6% and 75.6% after 72 h, respectively (Figure 2G–I).

### 3.5. Cells Migration

PPE and MPE impact on DASC, T98G and LN-18 cells migration was assessed using an in vitro scratch wound assay. Images of scratch areas from the time points 0, 12 and 42 and the percentage of the open wound area are illustrated in Figure 3. Our data show that MPE inhibited cell migration more strongly than PPE in the DASC (to 33.6% after 42 h) and LN-18 (to 26.0% after 42 h) cell lines. Regarding the T98G cell lines, both MPE and PPE inhibited cell migration to a similar extent (to 27.0%).

### 3.6. Cell Cycle

The effects of PPE (30 μg/mL) and MPE (30 μg/mL) on the cycle of DASC, LN-18, and T98G cells after 48 h are illustrated in Figure 4. Our data demonstrate that PPE induced cell cycle arrest in the subG1/G1 phase in T98G (increased to 24.0% ± 2.6) and LN-18 (increased to 11.7% ± 0.6) cells compared to control (*p* < 0.05), but not in DASC cells. MPE induced cell cycle arrest in the subG1/G1 phase only in the T98G cell line (increased to 15.8% ± 0.5). The changes in DASC cell cycles were not observed.

### 3.7. Cell Apoptosis

In our study, we examined the impact of PPE and MPE treatment on glioma cell apoptosis (DASC, LN-18, T98G) by annexin V and PI staining. The results (Figure 5) showed that PPE caused increased early apoptosis (lower right quadrat) in DASC, T98G, and LN-18 cells (by approximately (75%, 38% and 77%, respectively, compared to control) and late apoptosis/necrosis (upper quadrat) in DASC and T98G cells (by 25% and 43%, respectively, compared to control). Treatment with MPE led to early apoptosis in DASC and LN-18 cells (by 60% and 74%, respectively, compared to control) and late apoptosis/necrosis in DASC and T98G cells (by 38% and 58%, respectively, compared to control).

## 4. Discussion

Propolis owes its complex chemical composition to the quality of the resinous materials gathered by honey bees from different floral sources available around the hive. The quality of the resins has an impact on the quality and bioactivity of propolis. The chemical composition of the tested propolis was characterized by a similar amount of the identified active components and the total content of phenols, which is consistent with the classification of propolis from New Zealand as the “Poplar” type. Kumazawa et al. [23] conducted a comparison of the antioxidant activity and composition, as well as total phenol and flavonoid content, of individual samples of ethanolic extracted propolis from 14 countries and showed that New Zealand-sourced propolis was similar in composition to propolis from Bulgaria, Uzbekistan, Hungary, and three South American countries: Chile, Uruguay and Argentina. In our analysis we found high content of compounds such as pinobanksin, pinobanksin 3-acetate pinocembrin, chrysin or galangin. These compounds are characteristic of propolis originating from bud exudates of *Populus nigra* [6,24]. The analysis also confirmed research results published by other authors who have demonstrated that New Zealand propolis has very high levels of pinocembrin and pinobanksin-3-O-acetate [4].

The TPC value in our study was on high level > 240 mg GAE/g in both propolis. Other authors detected varying amounts of TPC in propolis. The values ranged from 14.6 to 150.8 mg GAE/g in Polish propolis [25] and from 99 ± 4.0 to 775 ± 8.5 mg GAE/g in Manuka propolis [26]. The TPC value often depends on the extraction method utilized.

Diffusion of heavy metals in the environment, occurring as a result of various human activities, results in penetration of these elements into food and direct human exposure to their toxic effects. Toxic elements such as, Cd, and Pb, even in trace amounts, present a risk to human health, causing non-communicable diseases with long-term effects. In our study, the level of As and Pb was higher in MPE than in PPE (Table 4). Comparing the obtained results with the Commission Regulation (EC) No 629/2008 [27] standards for supplements (Pb, 3.0 mg/kg; Cd, 1.0 mg/kg), we found that the level of Pb (3.74 mg/kg) in MPE was exceeded, but the level of elements was assessed in the lyophilizates. The obtained Pb content in the lyophilized extract was recalculated to Pb content in the liquid extract (0.935 mg/kg) and did not exceed the standards. Polish propolis was also analyzed by Matuszewska et al. [28]. The concentrations of As, Cd and Pb (As, 0.07 mg/kg; Cd, 0.04 mg/kg; Pb, 0.64 mg/kg) were higher than in PPE but not MPE. High concentration of Pb (5.74 mg/kg), Cd (0.194 mg/kg), and As (0.657 mg/kg) in Polish propolis was also indicate Roman et al. [29]. It should be noted that our PPE was obtained from green areas, free from pollution (Podlasie region), while propolis analyzed by Roman et al. [29] from the urban regions. Scientific research confirms that high amounts of toxic elements in propolis may result from the level of urbanization of a region. Therefore, the content of these elements in propolis should be constantly monitored. Studies by Ahamed et al. [30] confirm that Pb can influence of viability, cell cycle, lipid peroxidation, and caspase activation in human lung epithelial (A549) cells. However, it should be noted that propolis is a product that contains a number of compounds with antioxidant potential. In a study by Mu et al. [31], the cell viability assay results indicated that three phenolic acids—chicoric acid, isochlorogenic acid C, and caffeic acid—alleviated the cytotoxicity induced by Pb2+.

Due to the presence of a large number of antioxidant substances, propolis exhibits powerful anticancer activity, which has been confirmed in many studies [12,13,14,32]. Our previous study has revealed that Polish propolis decreases viability and has an antiproliferative activity and additionally, synergistically cooperates with temozolomide (TMZ), enhancing its growth-inhibiting activity against U87MG glioblastoma cell line through the reduction in NF-κB activity [11]. Catchpole et al. [4] have demonstrated that propolis from New Zealand has a strong antiproliferative effect against gastro-intestinal cancer cells DLD-1, HCT-116, KYSE-30, and NCI-N87, due to the high level of phenolic compounds (pinocembrin, pinobanksin-3-O-acetate and others). Propolis from Brazil has been demonstrated to exert a strong inhibitory effect on cell growth in glioblastoma (U251 and U343) and fibroblast cell lines (MRC5), although not on apoptosis, demonstrating a cytostatic action [33]. In this study, comparing the effect of both propolis (PPE and MPE) extracts on different glioma cell lines, we found strong, decreasing viability and antiproliferative effects on DASC, T98G and LN-18 cells (Figure 1 and Figure 2). Moreover, in scratch assay, it has been showed that PPE and MPE inhibit cell migration (Figure 3). Another study also confirmed the anti-migratory potential of propolis in cancer cells. Chang et al. [34] showed that treatment with different concentrations of Chinese propolis (25, 50 and, 100 μg/mL) and CAPE (25 μg/mL) significantly inhibited the proliferation and migration of the LPS-stimulated MDA-MB-231 breast cell line. Begnini et al. [35] reported that Brazilian Red Propolis (25 and 50 μg/mL) strongly inhibited migration in human bladder cancer 5637 cells.

Propolis shows anti-cancer activity through a various mechanisms. In our study, we examined the influence of MPE and PPE on cell cycle and apoptosis in the DASC, LN-18, and T98G cell lines. Both propolis extracts induced cell cycle arrest in T98G in the subG1/G1 phase, but PPE only in LN-18 cell line. Lack of cell cycle changes in DASC can be associated with a low proliferation capacity of that cells (Figure 4). What is more, PPE and MPE may induce cell necrosis, especially in T98G and DASC cells (Figure 5). Frión-Herrera et al. [36] showed that Cuban red propolis induced mitochondrial dysfunction and LDH release in breast cancer cell line (MDA MB-231), which indicated cell necrosis associated with reactive oxygen species production and decreased cell migration. The accumulation of cell population in the Sub-G1 phase may suggest that propolis did induce apoptosis. Interestingly, we also observed that PPE and MPE treatment induced cell cycle arrest in the S phase in T98G cells (*p* < 0.05). Other authors have also observed this effect. Jiang et al. [37] reported that Special Chinese propolis sourced from the Changbai Mountains showed anti-proliferation activity in SGC-7901 human gastric cancer cells by inducing both death receptor-induced apoptosis and mitochondria-mediated apoptosis, as well as cell cycle arrest in the S-phase. In this study, it has been observed that both PPE and MPE induced apoptosis in each glioma cell line (DASC, LN-18, T98G) (Figure 5). Zeynep et al. [38] also confirmed the apoptotic activity of propolis in C6 glioma cells. They showed that an ethanolic extract of propolis induced apoptosis in C6 glioma cells by activating the caspase cascade pathway, increasing caspase-8, -9, and -3 expression levels. The study by Noureddine et al. [39] showed that Lebanese propolis induced an increase in SubG0 fraction in Jurkat, glioblastoma (U251) and breast cancer (MDA-MB-231) cells. This increase in SubG0 was further investigated in Jurkat cells by annexinV/PI and showed an increase in the percentage of cells in early and late apoptosis as well as necrosis.

Many publications have explored significant anti-cancer properties of individual components of propolis. Szliszka and Krol [40] suggested that polyphenols from propolis sensitized tumor cells to TRAIL-induced apoptosis. The compounds, in combination with TRAIL, exhibit a strong cytotoxic effect on cancer cells [41,42]. Caffeic acid phenethyl ester (CAPE) inhibits NF-kB and enhanced the extrinsic pathway of apoptosis in cancer cells induced by TRAIL and Fas receptor stimulation [43]. The most recent research has demonstrated that CAPE displays significant cytotoxicity towards two glioma cell lines: Hs683 and LN319 [44]. Other authors have also confirmed that CAPE exhibits powerful antitumor effects on the following cancer cells: fibroblasts from oral submucous fibrosis (OSF), neck metastasis of Gingiva carcinoma (GNM) and tongue squamous cell carcinoma (TSCCa) [45]. Chrysin shows antiproliferative activity against human colorectal cancer cell line HCT-116, liver cancer cell line HepG2 and nasopharyngeal line CNE-1 to TNF-α-induced apoptosis [46]. Chrysin induces apoptosis in cancer cells by the activation of caspases and suppression of anti-apoptotic proteins such as IAP, c-FLIP, PI3K/Akt signal pathway, inhibition of IKK and NF-kB activity [47].

In this study, the potential activity of propolis extract against glioma cells was demonstrated, and the quality and safety of propolis were considered. Different glioma lines and astrocytoma cell line were used to compare whether the direction of action of propolis may be similar in different types of glioma. These are preliminary studies conducted in vitro.

Key factors in assessing a propolis extract, as well as other natural products in glioma treatment, are its bioavailability, metabolism, active compounds, and blood–brain barrier (BBB) permeability [48]. Bioavailability of some propolis compounds, such as flavonoids (chrysin and galangin), is low, and they are rapidly metabolized, which may limit the therapeutic potential of propolis extracts [49,50]. Despite that, Curti et al. [50] showed that oral uptake of brown propolis is followed by rapid metabolism and by cellular adaptation through the modulation of the concentration of first line antioxidant enzymes (SOD-1). Moreover, it cannot be excluded that the activity of propolis depends on synergisms between polyphenols and other active compounds [50]. An effective antiglioma agent must cross the BBB. BBB is permeable to some phenolic acids, such as caffeic acid (presented in studied extracts) [51] and flavonoids such as naringin, quercetin, genistein, epigallocatechin or its metabolites, but is neglected by some others, such as resveratrol and curcumin [52,53]. Future investigations including in vivo studies with cyclic administration of propolis, examination bioavailability and BBB permeability of propolis compounds are necessary for the evaluation of the therapeutic potential of propolis extracts.

## 5. Conclusions

In summary, the above results show that propolis from Poland and propolis from New Zealand (Manuka) have antiproliferative and pro-apoptotic activity on the human diffuse astrocytoma cell line (DASC) (Grade II glioma) derived from a patient and glioblastoma multiforme T98G and LN-18 cell lines from the ATCC. The anticancer potential was confirmed through induction of apoptosis, cell cycle arrest on subG1/G1 and S phase and attenuate migration. The PPE and MPE activity may be associated with the high content of antioxidant compounds in both types of propolis. The chemical composition of both propolis was comparable, with marginal differences in the amount of some compounds. The content of As and Pb in MPE was higher than in PPE.

In conclusion, Polish and New Zealand propolis extracts showed anti-glioma activity in in vitro study. However, further in vivo studies are required to confirm the therapeutic potential of propolis.

## Figures and Tables

**Figure 1 antioxidants-11-01305-f001:**
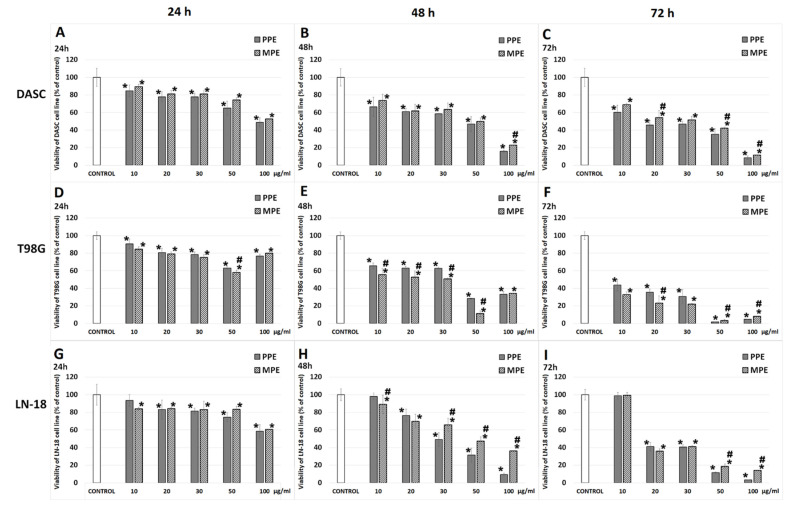
The viability of DASC (**A**–**C**), T98G (**D**–**F**) and LN-18 (**G**–**I**) cells after treatment with PPE and MPE (in concentrations 10, 20, 30, 50, 100 μg/mL) after 24, 48 and 72 h of incubation. The results are presented as a percentage of control. All statistical analyses were performed using Student’s *t*-tests or Mann–Whitney U tests (significant changes: * *p* < 0.05 vs. control, # PPE vs. MPE).

**Figure 2 antioxidants-11-01305-f002:**
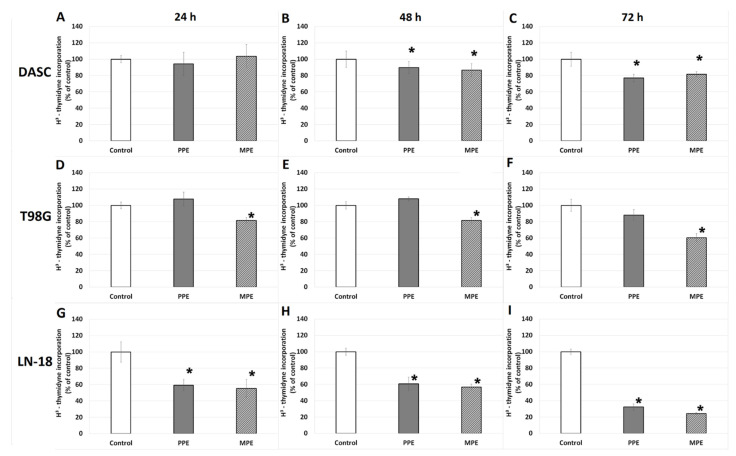
The [^3^H]-thymidine incorporation into DASC, T98G and LN-18 cells after treatment with PPE and MPE. Legend: [^3^H]-thymidine incorporation into DASC (**A**–**C**) and T98G (**D**–**F**) and LN-18 (**G**–**I**) cells after 24, 48, 72 h of incubation with PPE and MPE (in concentrations 30 µg/mL). The results are presented as a percentage of control. All statistical analyses were performed using Student’s *t*-test (significant changes: * *p* < 0.05 vs. control).

**Figure 3 antioxidants-11-01305-f003:**
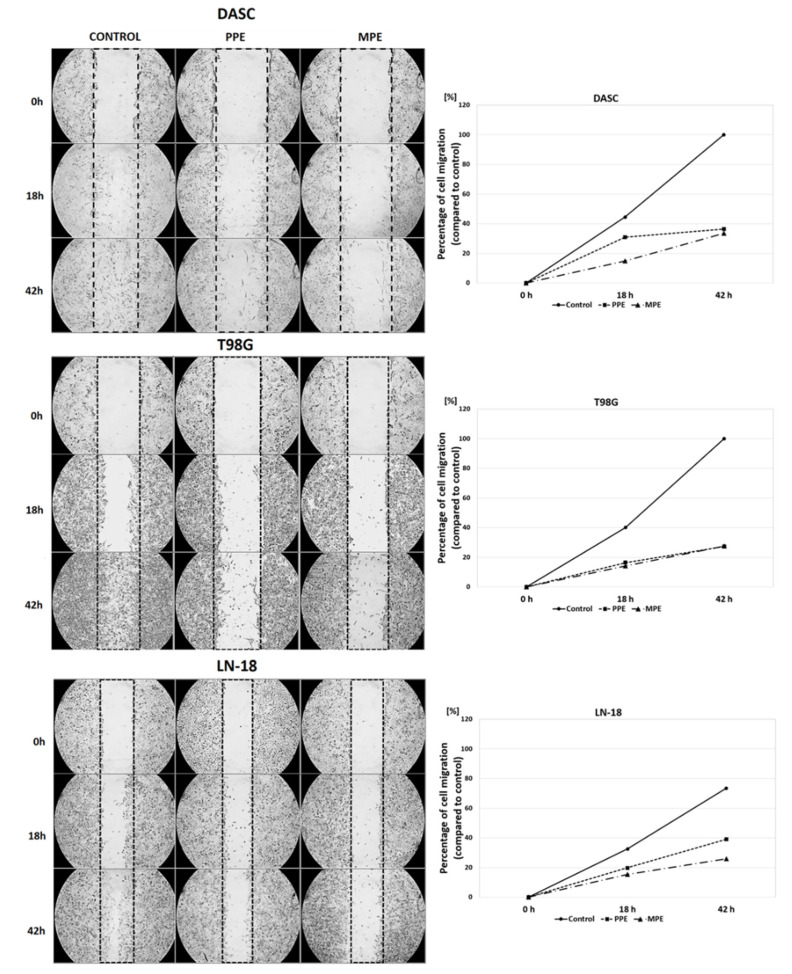
The effects of EEP (6.25 μg/mL) and MPE (30 μg/mL) on DASC, T98G and LN-18 on cells migration after 0, 18, 42 h of incubation.

**Figure 4 antioxidants-11-01305-f004:**
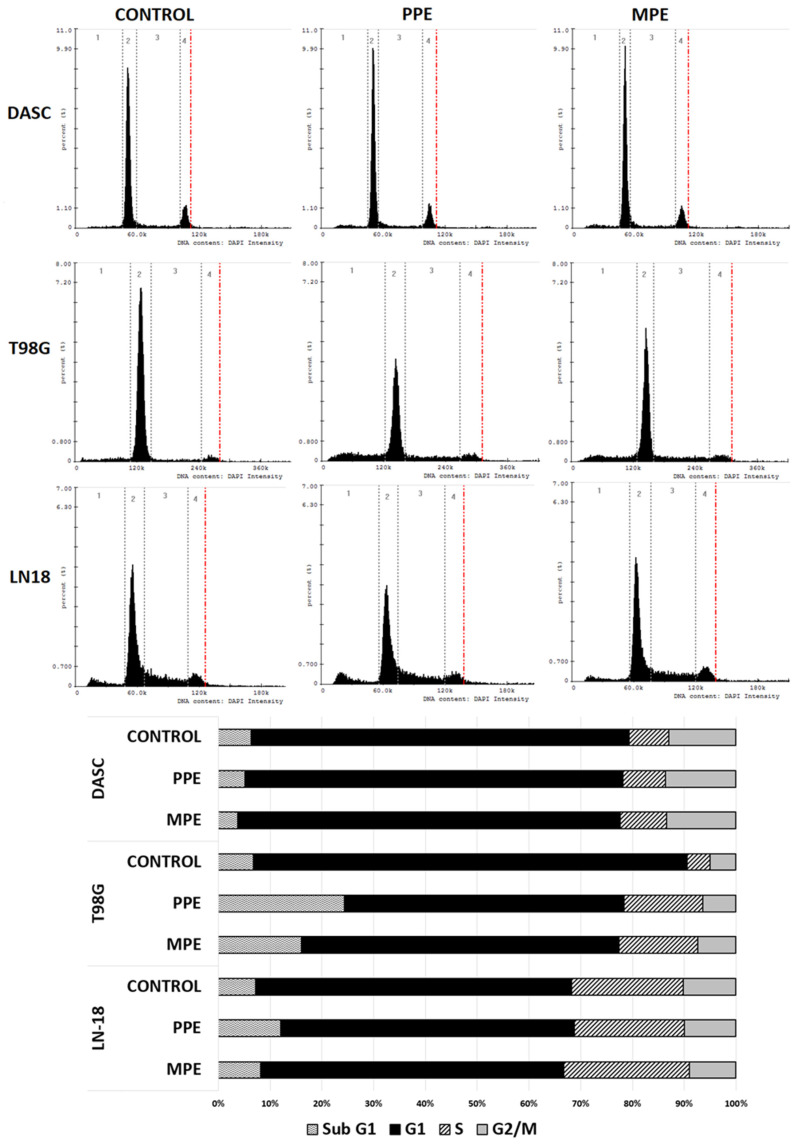
The effect of PPE and MPE on cell cycle analysis. DASC, T98G and LN-18 cells were incubated for 48 h with PPE (30 µg/mL) and MPE (30 µg/mL). Both the histogram and the bars present distributions of cells in subG1, G1, S and G2/M phases of the cell cycle.

**Figure 5 antioxidants-11-01305-f005:**
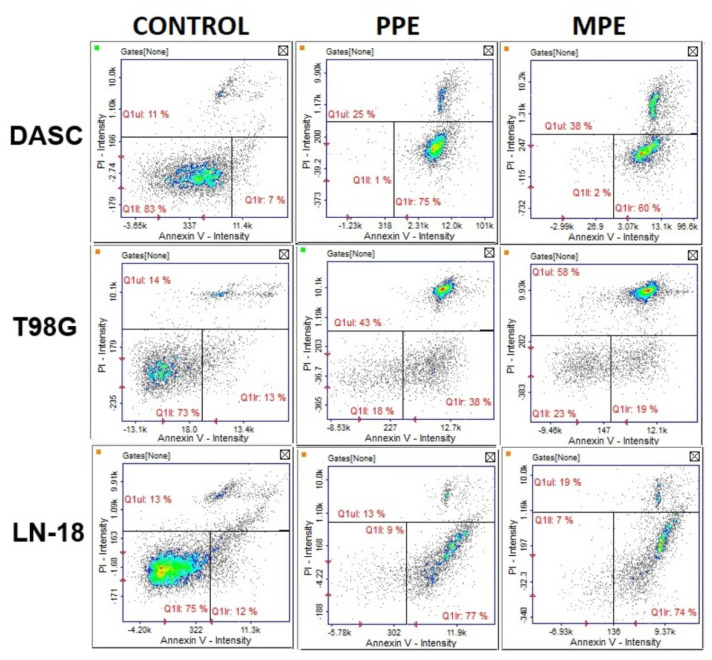
The quantitative assessment of DASC, T98G and LN–18 cells apoptosis induced by PPE and MPE (30 µg/mL) using Annexin V/PI staining.

**Table 1 antioxidants-11-01305-t001:** Results obtained in the quality control process.

Element	Precision (%)	Recovery (%)	Declared Concentration in CRM (µg/kg)
As	3.3	99.0	10
Cd	2.5	99.1	7
Pb	2.4	99.5	52

CRM—certified reference material.

**Table 2 antioxidants-11-01305-t002:** The main chemical compounds of the ethanolic extracts of propolis from Poland (PPE) and propolis from New Zealand (MPE), determined by GC-MS.

Components, TMS Derivative	*I* _T_ ^Exp^	*I* _T_ ^Lit^	PPE [%]	MPE [%]
Benzoic acid	1244	1247	1.80	0.33
Cinnamic acid	1542	1546	0.20	1.82
p-Coumaric acid	1944	1947	9.77	0.87
3,4-Dimethoxycinnamic acid	2030	2034	-	1.51
(E)-Ferulic acid	2101	2101	3.22	0.15
(E)-Caffeic acid	2155	2155	2.10	1.53
3-Methyl-3-butenyl (E)-caffeate	2371	2367	1.18	3.39
3-Methyl-1-butenyl (E)-caffeate	2374	2375	0.50	0.44
3-Methyl-2-bytenyl (E)-caffeate	2425	2421	1.65	2.36
Pinocembrin, mono-TMS	2460	2461	1.14	0.46
Benzyl (E)-p-coumarate	2516	2515	3.78	0.37
Pinocembrin	2551	2552	6.93	14.10
2-Phenylethyl p-coumarate	2603	2603	1.02	0.11
Pinobanksin	2613	2611	4.25	4.73
Pinobanksin 3-acetate, mono-TMS	2634	2632	1.26	0.21
Chrysin, mono-TMS	2655	2648	1.95	0.42
5,7-Dihydroxy-3-methoxyflavanone	2675	2673	2.02	2.04
Benzyl (E)-ferulate	2680	2680	1.64	0.45
Pinobanksin 3-acetate, di-TMS	2694	2693	10.01	9.00
Benzyl (E)-caffeate	2723	2722	3.79	2.70
Chrysin, di-TMS	2746	2745	5.33	5.73
Galangin, tri-TMS	2767	2769	8.95	9.60
CAPE	2805	2805	1.29	1.15
Cinnamyl (E)-p-coumarate	2836	2833	1.91	0.23
Sakuranetin	2877	2880	0.55	0.05
Quercetine	3218	3213	0.11	-

**Table 3 antioxidants-11-01305-t003:** Group composition of ethanolic extracts from Poland (PPE) and New Zealand (MPE) propolis.

Group of Compounds	PPE [%]	MPE [%]
Flavonoids and chalcones	49.4	52.1
Aromatic acids	18.3	7.8
Cinnamic acid esters	19.8	14.5
Phenylpropenoid glicerydes	1.3	0.0
Aliphatic and aromatic alcohol	0.2	0.8
Aliphatic acids	0.8	0.2
Carbohydrates	6.2	18.7
Sesquiterpenoids	0.0	0.2
Other compounds	4.0	5.7
Total	100.0	100.0

**Table 4 antioxidants-11-01305-t004:** Total phenolic content and toxic elements concentration of ethanolic extract from Poland (PPE) and New Zealand (MPE) propolis.

Extracts	TPC [mg GAE/g] Mean ± SD	Toxic Elements [mg/kg]		
		As	Cd	Pb
PPE	243.7 ± 9.0	0.00	0.01	0.16
MPE	245.6 ± 5.9	0.88	0.01	3.74

## Data Availability

Data is contained within the article and Appendix A.

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
