# Peer review of "Polish and New Zealand Propolis as Sources of Antioxidant Compounds Inhibit Glioblastoma (T98G, LN-18) Cell Lines and Astrocytoma Cells Derived from Patient"

_antioxidants, 2022, doi:10.3390/antiox11071305_

Round 1
Reviewer 1 Report
The MS of Moswa et al. investigates the biological effects of propolis on malignant glial cell lines. The MS has a two fundamental problems. a) Authors perform experiments on three malignant cell types. There are no primary (healthy) astrocytes and or neurons. In the absence of appropriate controls the results can not be properly evaluated. b) Authors do not discuss the major problem of CNS drug research, the permeability of blood brain barrier.
A possible experiment approach could be: Using nude mice, injecting tumor cells into the brain, adding propolis per os, or intraperitoneally.
Reviewer 2 Report
The manuscript is quite straightforward, but there is some confusion in its presentation. As for example, many paragraphs in the Results section should be inserted in a specific Discussion section to be added, otherwise there is a confused alternance between original data and bibliography data. Besides, with this structure, the relevance of previous studies and the meaning of the study in this context is difficult to catch. In the Discussion/Conclusion, some hints should be provided about studies in which the anti-cancer effect of propolis was tested in models other than cell lines. The different behaviour towards the 3 cell lines should be also discussed more accurately.
A few points
In The Abstract, it is mentioned that the experimental plan was to test propolis against glioblastoma. However, propolis was also tested against astrocytoma cells from a patient: this makes the focus of the manuscript confused, a clearer explanation of the astrocytoma cell model should be provided
Line 255-256 the sentence “Comparison of the chemical composition of the tested propolis revealed that both PPE and MPE had similar quantities of the identified active components” suggest that samples were statistically compare, but the statistical methodology is unclear. In addition, lines 255-263 are more appropriate for the Discussion section. The same for lines 281-294, lines 368, 377, lines 383-388
At line 279 authors wrote that “Pb content in terms of the liquid extract was 0.935 mg/kg and did not exceed the standards.”, but it is unclear whether this is an original data (in this case, according to what methodology? why the data was not provided in the Table?) or it is a data based on bibliography (in this case, what was the bibliographic source?)
The role of antioxidants in cancer biology (line 304) is far from being fully elucidated and should be introduced considering its complexity (see for example Schafer, Z., Grassian, A., Song, L. et al. Antioxidant and oncogene rescue of metabolic defects caused by loss of matrix attachment. https://doi.org/10.1038/nature08268), In addition, lines 303-307 are more appropriate for the Introduction section, as well as lines
Please explain the meaning of “reflection effect” (line 330). In addition, how could authors explain the evidence that this effect was cell-type specific and not compound-specific?
Line 401-406 are confused: What do authors means for “What is more PPE and MPE may induced cell necrosis.” Is it referred to their observations or to bibliographic data?
Lines 407-416 appears pointless and unnecessary, as no assessment on the activation of Chk1 and Chk2 kinases by sensory kinases ATR and ATM was carried out.
In the Conclusions, author suggests that “Polish and New Zealand propolis extracts have a promising anti-glioma potential”. Authors should add what kind of study would be necessary to confirm the anti-glioma potential
Minor points
In the Abstract, please avoid colon after “such as”
Please provide the acronym of TZM at line 36
Please when stating “In this study, more than 100 individual compounds in PPE and more than 150 compounds in MPE were identified.” mention the methodology used to obtain this result
In the Introduction “both propolis characterized a lot of antioxidants compounds” means “both propolis contained a lot of antioxidants compounds”?
Line 301. Please avoid the term “to immunize”, because it suggests an immune reaction, but the role is more properly to maintain germ-free or germ-low content
Line 314 Please specify how cell viability was determined
In Figure 1 and 2, please indicate clearly in the panels 24, 48 and 72 hrs (vertical) and cell type (horizontal), without repeating cell type 3 times
Line 453 Please provide the acronym for CAPE
Round 2
Reviewer 2 Report
The manuscript appears clearer, but a few points can be still improved. Authors wrote that in their study, after applying a dose of propolis higher than 50 ug/ml, the viability of T98G cells increased in a cell- dependent manner “reflection effect”. Could they discuss what could be the reason underlying this effect? In addition, authors replied that further study are required to confirm therapeutic potential of propolis, but they do not suggest what kind of studies would be necessary
Minor points:
- Line 58: Please correct “A research studies”
